# Analytical Prediction of Coal Spontaneous Combustion Tendency: Pore Structure and Air Permeability

**Bin Du** [1,2,3,*], **Yuntao Liang** [1,2,3], **Fuchao Tian** [1,2,3,*] **and Baolong Guo** [2,4]

1 China Coal Science Research Institute, Beijing 100013, China
2 State Key Laboratory of Coal Mine Safety Technology, China Coal Technology and Engineering Group, Shenyang Research Institute, Shenfu Demonstration Zone, Fushun 113122, China
3 College of Emergency Management and Safety Engineering, China University of Mining and Technology (Beijing), Beijing 100080, China
4 College of Mining and Safety Engineering, Shandong University of Science and Technology, Qingdao 266590, China
* Correspondence: dubin1127@126.com (B.D.); tianfuchao@cumt.edu.cn (F.T.)

**Abstract:** In previous research, many scientists and researchers have carried out related studies about the spontaneous combustion of coal at both the micro and the macro scales. However, the macroscale study of coal clusters and piles cannot reveal the nature of oxidation and combustion, and the mesoscale study of coal molecule and functional groups cannot be directly applied to engineering practice. According to our literature survey, coal is a porous medium and its spontaneous combustion is a multi-scale process. Thus, the mesoscale study of coal's spontaneous combustion is essential. In this manuscript, the mesoscale of the coal body (such as pore size, pore volume, and specific surface area), and the meso-scale structural morphological characteristics of the coal surface are finely analyzed and characterized. On this basis, the meso-scale structure of pores and fractures are digitally reconstructed. Furthermore, velocity and pressure distributions of the flow field in the pores of the scan plane are outlined and described by numerical simulation. The results indicate that, because of the pore structure characteristics and fluid viscosity, not all fluids in the pores demonstrate flow. This conclusion well explains the source of CO gas in methane extraction pipes, which is one of the main index/indicator gases of the spontaneous combustion of coal.

**Keywords:** mesoscale; pore structure; fire risk assessment; coal spontaneous combustion; ignition tendency

## 1. Introduction

The demand for energy has grown with the boom in the global economy. Although clean energy has developed rapidly in recent years, coal still occupies an important position in the global energy landscape. However, fire accidents associated with the coal spontaneous combustion (CSC) caused by oxidation reactions during mining, transportation, and storage still occur in various countries of the world [1–4]. In the process of underground mining especially (Figure 1), the spontaneous combustion of residual coal in goafs occasionally happens. This phenomenon can be explained along two aspects: At the microscale, a series of coal–oxygen reactions of the molecules and the functional groups occur along the complex interfaces of the pores and fractures in coal [5,6], as a typical porous medium, which then release large amounts of reaction heat. At the macroscale, the coal left in goafs piles up in a loose and broken state, which facilitates the accumulation of heat released from coal oxidation, thus triggering the self-heating of the coal body.

To prevent the oxidation and spontaneous combustion of residual coal in goafs, experts and scholars all over the world have carried out extensive research work, as shown in Table 1.

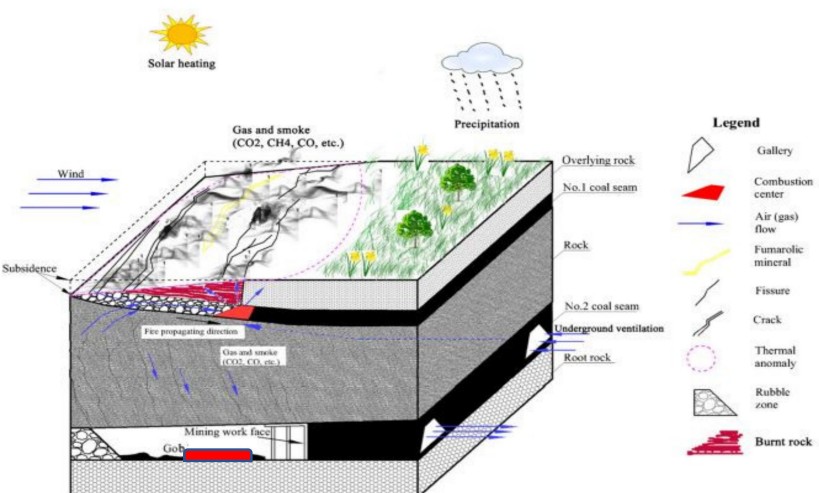

**Figure 1.** Illustration of an underground coal mine and fires in goafs [7].

**Table 1.** Representative research of CSC carried out by experts and scholars.

| Scale of Research | Perspective of Research | Main Research Content/Conclusions | Scientists and Researchers |
|---|---|---|---|
| Macroscale aspects | ①Temperature; ②Oxygen concentration; Oxygen adsorption ③Indicator gases; ④Air leakage; ⑤Particle size | Relationship between air leakage velocity range and high possibility of self-ignition | Lin et al. [8] |
| | | The conducting of a simulation study on the dynamic evolution law of CSC in high-temperature goafs | Wang et al. [9] |
| | | Investigation of the macro-kinetics of coal–oxygen reactions under varying oxygen concentrations | Yang and Li [10] |
| | | Consideration of underground coal fires as a typical combustion mode of smoldering during which the reaction zone is not exposed to a constant oxygen concentration | Yang and Li [10] |
| | | Investigation of oxidation products and indicator gases under different oxygen concentration conditions in the temperature range of 30–210 °C | Rychter and Smolinski [11] |
| | | Computational analysis of the influence of particle size, oxygen concentration, and furnace temperature on the ignition characteristics | M.P.R. [12] |
| Mesoscale aspects | ①Coal molecular structure; ②Functional groups; ③Free radicals; ④Activation energy; ⑤Some chemical inhibition methods; | Exploration of the changes of functional groups during low-temperature oxidation of coal by building a model compound of coal-like molecular structure. | Tang et al. [13–16] |
| | | Exploration of the relationship between exothermic kinetic properties of oxidation of key active groups and different metamorphic degrees | Deng [17] |
| | | Investigation of the correlations among free radicals, apparent activation energy, and functional groups during low-temperature oxidation | Wang [18] |
| | | Pre-oxidation treatment for functional groups with high reactivity to reduce spontaneous combustion risk | Li et al. [19] |
| | | Qualitative and quantitative analysis of the physical and chemical inhibition effects of the halogen salt inhibitor on active functional groups | Guo et al. [20] |
| | | Effects of scavenger (TEMPO) and ethylenediaminetetraacetic acid (EDTA) used to inhibit the chain reaction of free radicals and functional groups | Li et al. [21,22] |

In summary, most existing studies have been conducted at the macroscale and the microscale, while there are few reports on coal–oxygen reactions at the mesoscale.

At the mesoscale aspect, similar research about flow in the pores of porous media have focused mainly on soil or stone rather than on coal. For example, Heejun Suk and Eungyu Park [23] proposed and developed a new numerical method to accurately and efficiently compute and simulate variably saturated flow in heterogeneous layed soil. On the other hand, research with a significant reference to flow in the pores of porous media,

have mainly focused on oil or moisture rather than on air. For example, Ravi Borana et al. [24] simulated the instability phenomenon of the double-phase flow of oil and water in a petroleum reservoir, and Wang et al. [25] investigated a water-phase seepage evolution model considering the mesoscale characteristics of pores and fissures in coal. What is more, Pan et al. [26] and Chen et al. [27] have investigated the moisture effects on methane storage, transport, and diffusion in coal.

In fact, coal is a porous medium with complex internal pores and a large specific surface area, and the coal–oxygen reaction primarily occurs on the gas–solid surfaces of these pores, as shown in Figure 2. Studies concerning the macroscale aspect cannot reveal the nature of the coal–oxygen interaction and studies concerning the mesoscale aspect cannot be directly applied to engineering practice. Only studies concerning the mesoscale aspect can combine the research results of these two aspects of the macro- and microscopic.

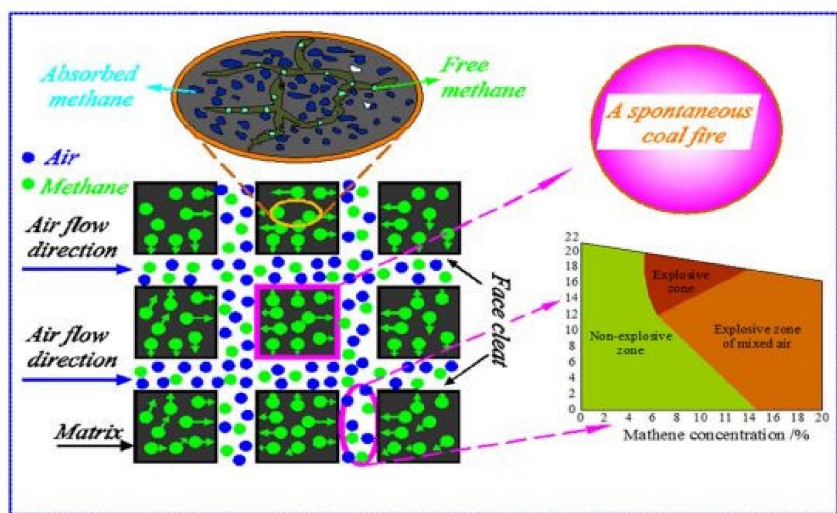

**Figure 2.** Mesoscale conceptual model of CSC [28].

Therefore, based on the basic theory of complex systems and on the research approaches of meso-science, Liang et al. [29–32] presented the concept of the mesoscale for coal spontaneous combustion. It was deemed that there exist five research scales in the mining space, the boundary scales among which include reactive particles, single coal particles, and stacked coal fields, while those between the boundary scales are intragranular pores (mesoscale I) and particle clusters (mesoscale II), as shown in Figure 3.

Therefore, it has been proposed to conduct cross-scale research on the CSC process using a correlation-type multi-scale method.

During the process of coal mine production, the mine ventilation can discharge toxic and harmful gases from the underground confined space and provide fresh air for underground workers. However, at the same time, it also provides continuous fresh air for coal–oxygen reactions, and thus provides one of the absolutely necessary conditions for coal oxidation and spontaneous combustion.

Thus, it is essential to conduct a study about the gas flow field and its mesoscale effects on CSC, especially regarding the effects of pore structural characteristics and air permeability on CSC, which makes an impact on the transport of oxygen molecules in porous media and thereby affects the oxidation reaction process of coal. This research facilitates CSC prevention and inhibition.

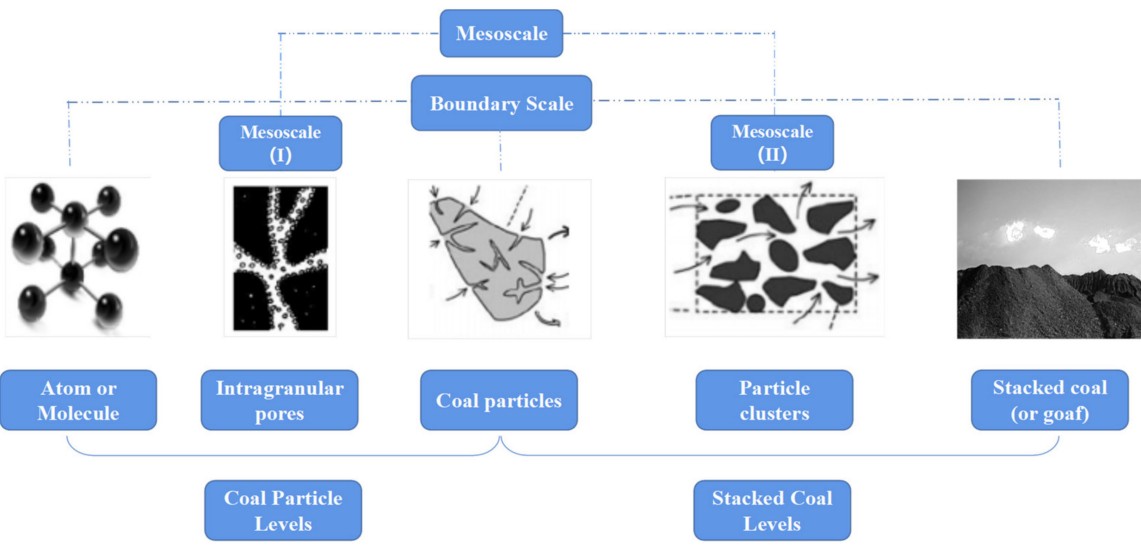

**Figure 3.** Five research scales and two mesoscales in the field of CSC.

## 2. Experiment Methods and Results

### 2.1. Determination of Pores and Specific Surface Area of Coal

2.1.1. Method

In this experiment, the pore structure and specific surface area of the coal sample were tested by an ASAP 2020 specific surface area and pore size analyzer (Figure 4) to explore the variation laws of its pore structure and distribution characteristics. First, about 2 g of the grinded coal sample collected in the Shuangma coal mine of Yinchuan city, in the Ningxia Hui Autonomous Region of China was weighed and put into the sample tube, and then the total weight of the tube and the coal sample was measured. Afterwards, liquid nitrogen was put into the cold trap. Before the experiment, the coal sample desorbed in advance in accordance with the following procedure. First, the sample tube was installed at the degassing station, and a heating package was put on the bottom. Next, the sample was heated to 100 °C at a heating rate of 5 °C/min and maintained for 100 min, then heated to 110 °C at a heating rate of 5 °C/min and maintained for 600 min, and finally allowed to naturally cool down to room temperature. After the above pretreatment, the experiment started. Specifically, a sample tube loaded with a coal sample was installed at the analysis station, and nitrogen was injected into it to measure the nitrogen adsorption capacity of the coal sample at −196 °C (nitrogen's critical temperature, 77 K). Based on the measured data, an adsorption curve was obtained to analyze the experimental results.

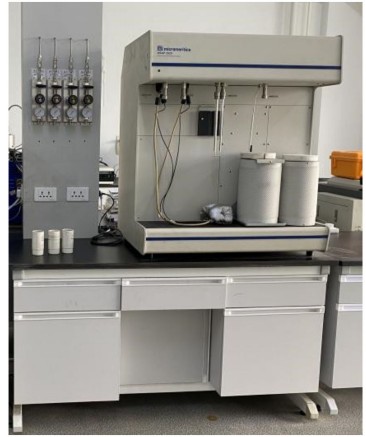

**Figure 4.** ASAP 2020 automatic physical and chemical adsorption instrument.

2.1.2. Results

According to the experimental results (Table 2 and Figure 5), the volumes of micropores, small pores, mesopores and macropores are 0.27 mm³/g, 4.62 mm³/g, 4.79 mm³/g and 1.55 mm³/g, respectively, their volume ratio being about 1: 17: 18: 6. The specific surface area of coal is 7.65 m²/g, which means that just 1 g of coal can provide a gas–solid reaction interface of 7.65 m² for the coal–oxygen reaction.

**Table 2.** Volumes of different-sized pores and specific surface area of coal.

| Pore Volume (mm³/g) | | | | | BET Specific Surface Area (m²/g) |
|---|---|---|---|---|---|
| **Micropore** | **Small Pore** | **Mesopore** | **Macropore** | **Total** | |
| 0.27 | 4.62 | 4.79 | 1.55 | 11.23 | 7.65 |

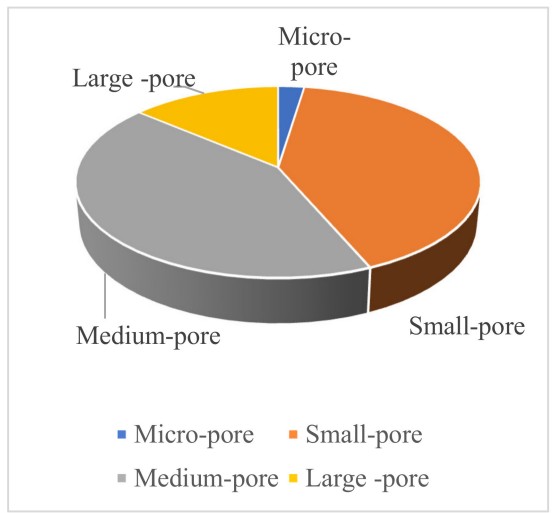

**Figure 5.** Volume distributions of different-sized pores.

*2.2. Scanning Electron Microscope (SEM) Test*

2.2.1. Method

The microscopic morphological characteristics of the pore fracture surface of a coal sample can be obtained with the aid of an SEM. To get an accurate understanding of the developmental characteristics of the internal pore structure of the coal body, the surface of the coal samples was scanned using a KYKY 2800B SEM (Figure 6). The internal pore structure was determined at an operating voltage of 25 kV (Figure 7).

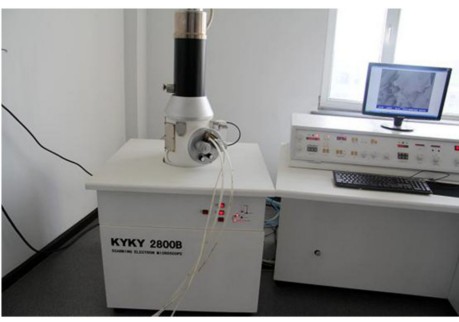

**Figure 6.** KYKY 2800B SEM.

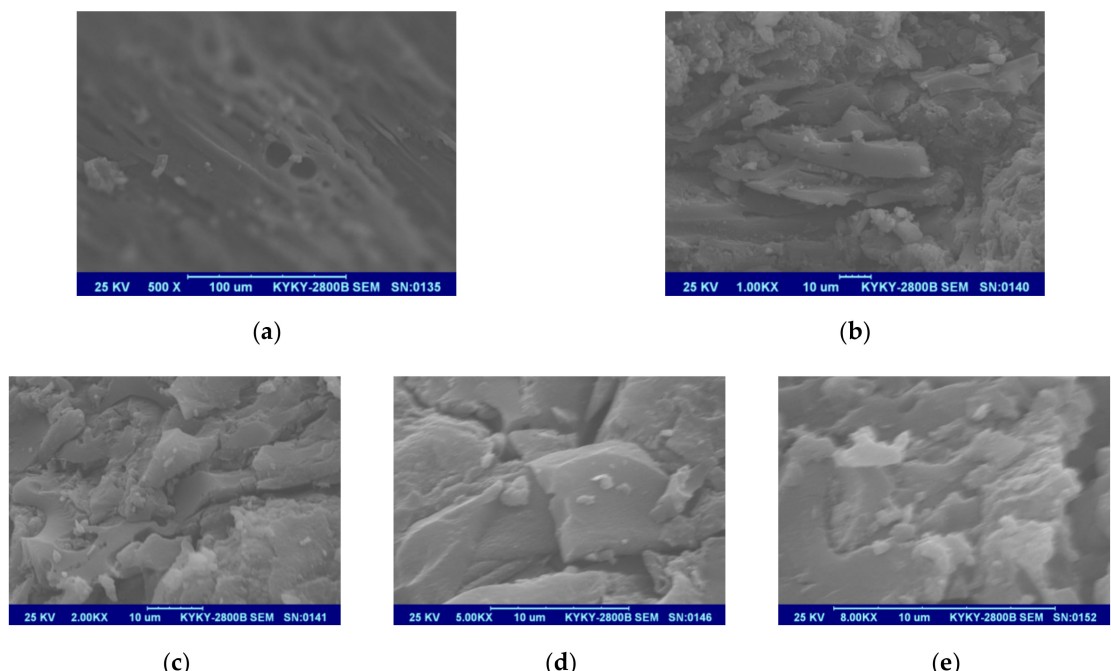

**Figure 7.** Microstructure morphological characteristics of the coal surface at different magnifications. (**a**) 500× magnification; (**b**) 1000× magnification; (**c**) 2000× magnification; (**d**) 5000× magnification; (**e**) 8000× magnification.

The coal sample was placed into the sample cell of an ion sputtering instrument and evenly plated with the metallic element on the surface with the help of a magnetic field. After 2–4 min of plating, the non-conductive charge phenomenon of the sample was eliminated, and the observation effect was meanwhile promoted.

The coal samples were then bonded to a conductive tape and placed into a field emission SEM sample chamber. Subsequently, the microscopic morphology of the coal surface was observed and subject to scanning imaging in an ascending order of magnification (500×, 1000×, 2000×, 5000×, and 8000×). In this manner, the microstructure morphological characteristics of the coal sample could be investigated through the test.

### 2.2.2. Results

At 500× magnification, pores inside the coal matrix, as well as their circular cross-sections, could be observed. This is attributed to the fact that coal is formed from plants. Over the long geological ages, these plants are metamorphosed at high temperatures and pressures, eventually forming coal with varying metamorphic degrees. The pores in coal may originate from vascular bundles that transport water or nutrients in plants.

At 1000–8000× magnification, the pore distribution is found to be diverse. The cross-sections of pores are mostly irregular in shape, which leads to a high flow resistance. The pores are well connected and rich in connecting channels, which is conducive to the flow of fluid within them.

### 2.3. Digital Reconstruction of Pore Structure

### 2.3.1. Method

Generally speaking, homogenized macroscopic methods are effective at simplifying the complex geometry of real porous materials when describing or characterizing the flow in the porous medium at the macroscale. The macroscopic methods to describe the behavior of pore space can be quantified using two average physical quantities, i.e., porosity or permeability. However, as can be seen from Figure 7, at the mesoscale the real pore structure is non-homogeneous and anisotropic in the coal body. Therefore, the use of

macroscopic physical quantities to simulate the mesoscale pore system may fail to reflect the real distribution of pores.

Arnab Kumar Pal et al. [33] have revealed the internal micro-structure and their connectivity across multiple scales with digital pore-scale images of sandstone. Joyce Schmatz et al. [34] imaged pore-scale fluid–fluid–solid contacts in sandstone with nanometer resolution using cryogenic broad ion-beam polishing in combination with scanning electron microscopy and phase identification by energy-dispersive X-ray analysis.

To reflect the real fluid flow in the pore structure at the mesoscale by the numerical simulation method, the SEM results of the coal sample surface magnified by 5000× were adopted to reconstruct the analytic structure of the internal pores of the coal sample at a scale of 1:1. With the aid of filtering and intensity slicing techniques, the raster image was digitized into vector data using vector software. In this way, a microscale pore structure model of the coal body could be extracted and then saved as a *.DXF file.

### 2.3.2. Results

In the SEM image, the dark-colored areas indicate pores and the light-colored areas indicate the matrix. In the digitally reconstructed analytic geometric model of the pore structure, the boundary between pores and the matrix is depicted with solid lines. In Figure 8, the closed solid lines indicate pores in coal, and it can be found that these pores are diverse in terms of shape and size. In addition, the boundaries of pores that are parallel to, and can flow within, the scan plane are depicted with solid carmine lines, while the boundaries of pores that are perpendicular to and do not flow in the scan plane are depicted with solid black lines.

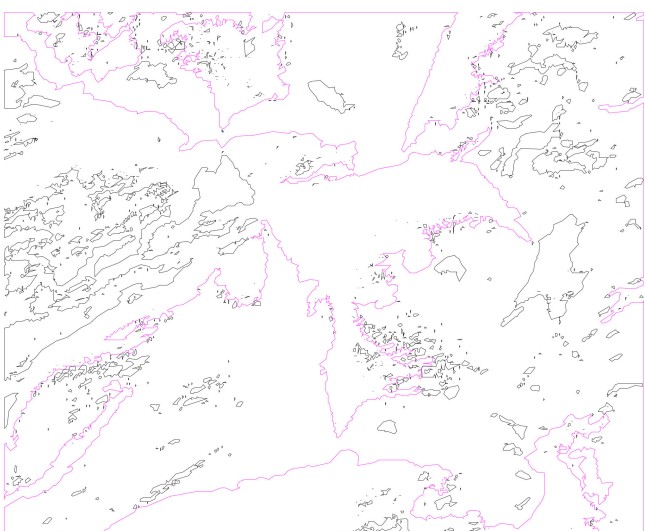

**Figure 8.** Results of digital reconstruction of the coal pore structure magnified by 5000X.

### 3. Numerical Simulation

#### 3.1. Mathematical Model

Due to the different research scales, the gas flow in coal pores no longer conforms to the N-S equation as the macroscale gas flow does.

(1)　N-S equation

$$\rho \frac{dv}{dt} = -\nabla p + \mu \Delta v + f$$

where $\rho$ is the fluid density, kg/m$^3$; $v$ is the velocity field; $p$ is the pressure, Pa; $\mu$ is the coefficient of viscosity, Pa·s; and $f$ is the body force, N.

(2)　Peristaltic flow

When the fluid flows in the microscale pore structure of coal, the viscous force, which is much larger than the inertial force due to the low Reynolds number, plays a dominant role, so the inertial force is completely negligible. Moreover, since the micro-geometric scale of the simulated object is at the nanometer level, the interface of the physical field is set to the peristaltic flow. The peristaltic flow, also known as the Stokes flow, is generally used to simulate flows with a low Reynolds number that occur in systems with a high viscosity or a small geometric scale, where the inertia term in the governing equation is negligible. The governing equation is:

$$\mu \Delta v = \nabla p$$

where $\mu$ is the coefficient of viscosity, Pa·s; $v$ is the velocity field; and $p$ is the pressure, Pa. In this paper, the gas flow in loose coal bodies and in the voids between them and the matrix, pores and fractures in the residual coal of a goaf was studied in a descending order of scale, starting from the macroscale.

### *3.2. Macroscale*

#### 3.2.1. Method

The research was carried out using a multi-physics field approach with the aid of the COMSOL numerical simulation software. Firstly, an area (300 m in length and 180 m in width) of stacked coal in a goaf was taken as the research object, where 0–40 m was the natural accumulation area, 40–100 m was the load-influenced area, and 100–300 m was the steady compaction area [35]. The widths of the air-inlet roadway and the air-return roadway were both 5 m. The rocks in a fall accumulation state and their voids in the goaf were regarded as the porous medium, and the relevant simulation parameters are listed in Table 3.

**Table 3.** Numerical simulation parameters [36].

| Parameter | Value | Description | Unit | Parameter | Value | Description | Unit |
|---|---|---|---|---|---|---|---|
| $M_a$ | 29 | Molar mass of air | - | $\kappa_1$ | $5.38 \times 10^{-6}$ | Permeability in the natural accumulation area | m$^2$ |
| $\varphi$ | 0.29 | Void ratio of goaf | - | $\kappa_2$ | $2.6 \times 10^{-6}$ | Permeability in the load-affected zone | m$^2$ |
| $\mu$ | $2.01 \times 10^{-5}$ | Dynamic viscosity | Pa·s | $\kappa_3$ | $1.3 \times 10^{-6}$ | Permeability in the compacted stable zone | m$^2$ |
| $D$ | $2.88 \times 10^{-2}$ | Gas diffusion coefficient | - | $L \times W$ | $300 \times 180$ | Goaf area | m$^2$ |

According to the actual situation, the airflow velocity at the inlet of the goaf was set to 0.9 m/s in the COMSOL Multiphysics numerical simulation software to analyze the overall distribution of gas flow velocity. After that, a two-dimensional section line was set in the middle of the goaf along the direction of the working face, so as to analyze the distribution of the air-leakage velocity at points on the line. The coordinates of the starting point of the line were (0, 80) and those of the ending point were (260, 80).

#### 3.2.2. Results

The overall distribution of the gas flow field in the goaf is presented in Figure 9. From Figure 9, it can be found that the air-leakage velocity of the goaf is high in the area of dense flow lines and low in the area of sparse flow lines. The area of dense flow lines is mainly located on the air-inlet side and the air-return side as well as the shallow part of the goaf.

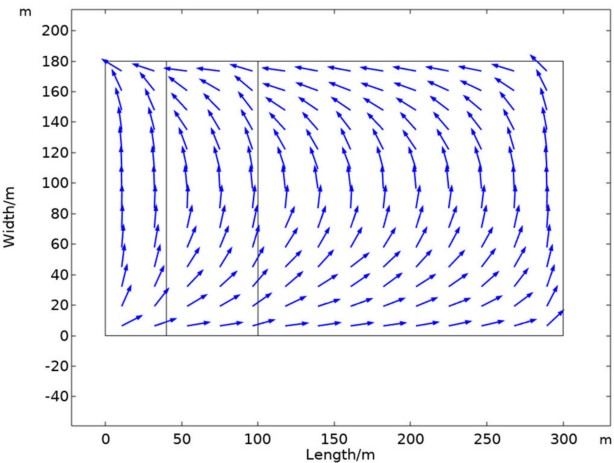

**Figure 9.** Distribution of gas flow field in the goaf.

The distribution of the air-leakage velocity in the goaf along the two-dimensional section line is given in Figure 10. It can be found that at the macroscale, the order of magnitude of the airflow velocity in goafs is "m/s".

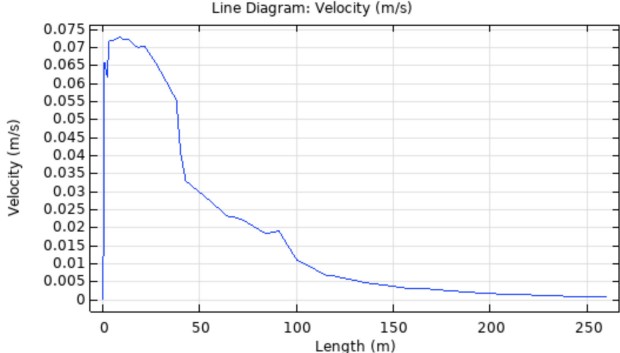

**Figure 10.** Curve of gas flow velocity along the two-dimensional section line in the goaf (y = 80 m).

*3.3. Mesoscale*

3.3.1. Numerical Simulation on Gas Flow in the Mesoscale Non-Analytic Pore Structure

Methods

To further investigate the particle clusters and the gas flow in the intragranular pores at the mesoscale, the residual coal in the oxidation heating zone (x = 40 m, y = 80 m) and its nearby area in the goaf were selected as the research object. The research area was 80 cm in length and 50 cm in height, and it was assumed that the coal blocks in it were oval in shape, with a long axis of 24 cm, a short axis of 8 cm, and an angle of 40° with the floor of the goaf. The porosity of the coal block was 9.5% and the permeability was $1 \times 10^{-16}$ m$^2$, as measured by experiments [37,38]. According to the correlation-type multiscale research method, as coal blocks were at the particle-agglomeration scale, their inlet airflow velocity was determined by their superior scale. From the airflow velocity distribution curve of stacked coal (Figure 10), it can be seen that the airflow velocity of the particle cluster was 0.04 m/s.

Results

The velocity field and pressure field distributions of the gas flow of the particle cluster and its nearby area were obtained through the simulation (Figures 11 and 12).

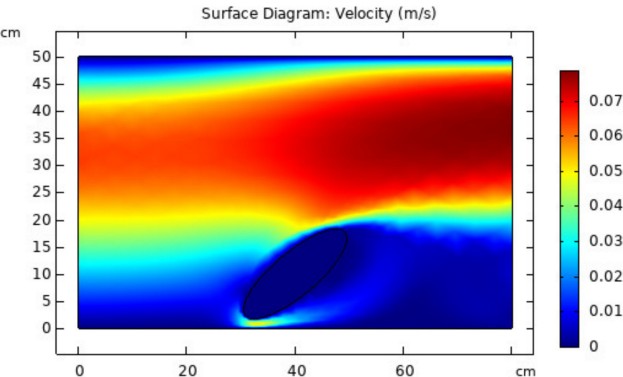

**Figure 11.** Distribution of gas velocity field near a particle cluster in the goaf.

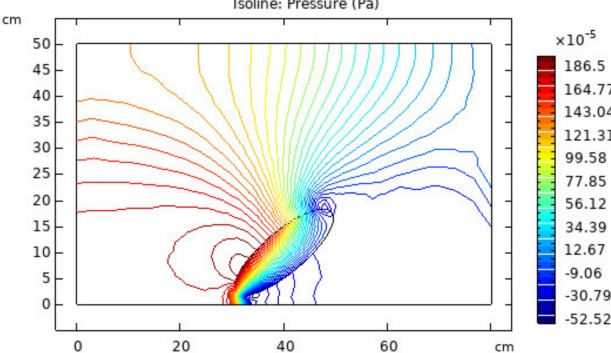

**Figure 12.** Distribution of gas pressure field near a particle cluster in the goaf.

(1)    Gas flow in the vertical direction at the mesoscale

To further study the gas flow inside intragranular pores in the vertical direction, a two-dimensional section line perpendicular to the floor and penetrating the particle cluster was set in Figure 13. The function expression is x = 40 cm (6 cm < y < 14.5 cm), which is marked by the red line segment in Figure 13.

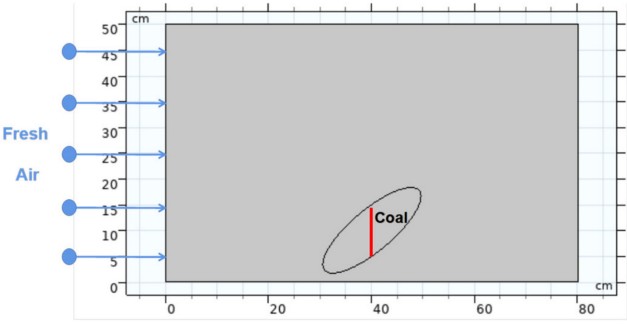

**Figure 13.** Location of the vertical two-dimensional section line in a particle cluster.

The distribution of the gas flow velocity in the particle cluster along the vertical two-dimensional section line was analyzed, and the simulation results are shown in Figure 14. From Figure 14, it is noticed that the closer to the surface of the particle cluster, the faster the gas flow in the intragranular pores.

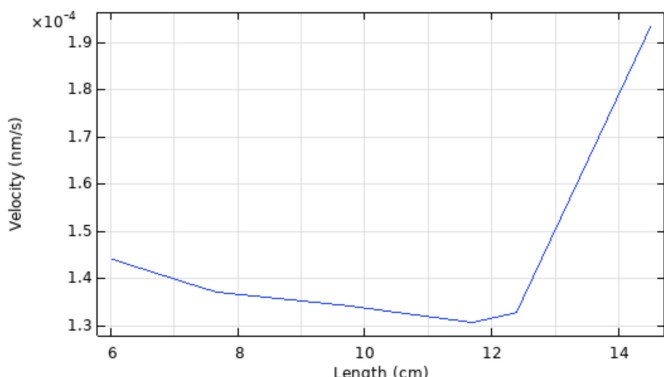

**Figure 14.** Distribution curve of gas flow velocity in pores along the vertical two-dimensional section line.

(2)    Gas flow in the horizontal direction at the mesoscale

To further study the gas flow inside intragranular pores in the horizontal direction, a two-dimensional section line parallel to the floor and penetrating the particle cluster was set in Figure 15. The function expression is y = 7 cm (32 cm < x < 42 cm), which is marked by the red line segment in Figure 15.

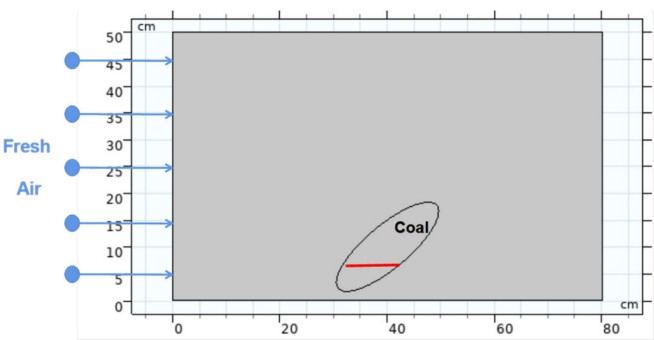

**Figure 15.** Location of the horizontal two-dimensional horizontal section line penetrating a particle cluster.

The distribution of the gas flow velocity in the particle cluster along the horizontal two-dimensional section line was analyzed, and the simulation results are shown in Figure 16. From Figure 16, it is observed that gas flows faster in the intragranular pores at positions closer to the windward side of the particle cluster.

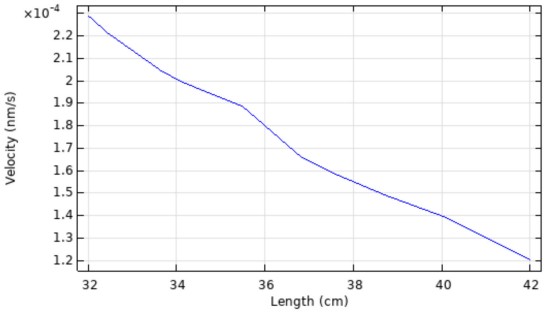

**Figure 16.** Distribution curve of gas flow velocity in pores along the horizontal two-dimensional section line.

(3)  Summary

The above analysis on the mesoscale gas flow velocity in pores within the particle cluster reveals that the gas flow velocity in intragranular pores is in the order of magnitude of "nm/s". This is at the same scale as the pore size of intragranular pores, which proves the correctness of the simulation results. In addition, it is observed that gas flows faster in the particle pores at positions closer to the surface and windward side of the particle cluster.

### 3.3.2. Numerical Simulation on Gas Flow in the Mesoscale Analytic Pore Structure

### Methods

The vector geometry file was imported into the COMSOL Multiphysics numerical simulation software. Meanwhile, the fluid-flow area was converted into a computational domain whose length and height were set to 20 μm and 14 μm, respectively. The fluid flowed into the left side of the computational domain and flowed out from the right side, and the velocity on the inflow side was set to be $5 \times 10^{-3}$ nm/s according to the previous simulation results. The density of the fluid was 1.29 kg/m$^3$, the viscosity of it was $1.84 \times 10^{-5}$ kg/(m·s), and the ambient temperature was 298.15 K (Figures 17 and 18).

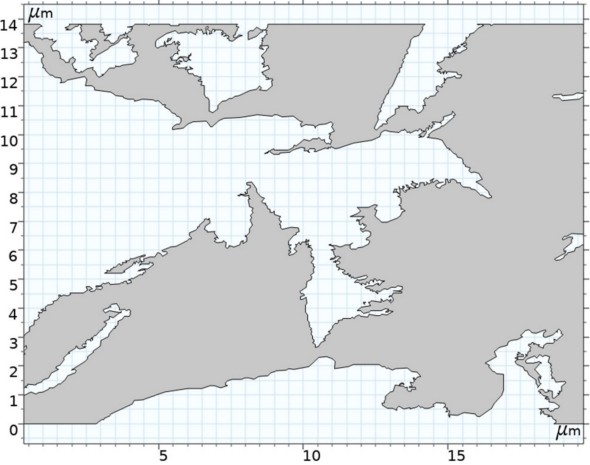

**Figure 17.** Reconstruction of the coal micropore structure model in COMSOL Multiphysics.

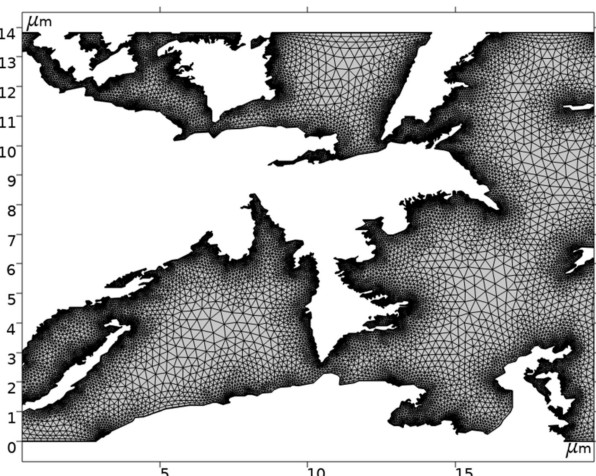

**Figure 18.** Diagram of grid division.

The computational domain was divided into grids using the physical field-controlled grid method.

Steady-state analysis was conducted on the distribution of the velocity and pressure fields after setting the geometric model, material parameters, physical field, and boundary conditions.

Results

After grid division, the model contains a total of 37,242 cells, including 996 vertex cells and 2603 boundary cells.

From the steady-state distribution of the velocity field (Figure 19), the fluid flows fast at the "throat" of the pore structure, and this flow gradually slows down as the pore diameter increases. Due to the influence of the pore structure characteristics and the fluid viscosity, not all of the fluids in pores are flowing. The fluid, which is able to find the best path, mainly flows along the shortest path between the inflow and outflow points. As the distance between the inflow and outflow points extends, the fluid no longer flows when the relative pressure difference between the two points becomes insufficient to offset the viscous resistance generated by the flow.

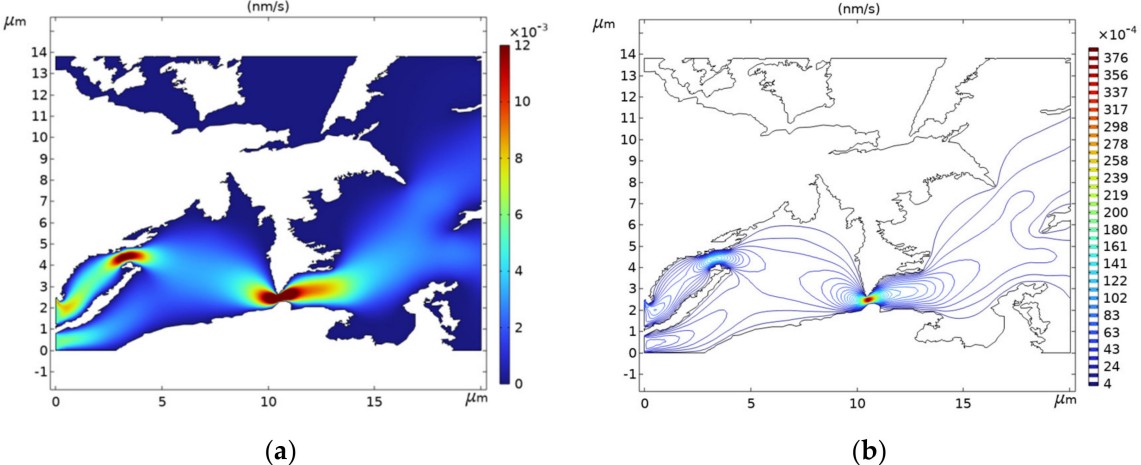

(**a**)  (**b**)

**Figure 19.** Distribution of the flow velocity field of mesoscale fluid with an analytic structure. (**a**) Surface map; (**b**) Contour map.

For a clear distinction between the flow area and the non-flow area, a flow line diagram of the flow field was plotted (Figure 20). It is found that the non-flow area is mainly located in small-sized pores, which is primarily attributed to the influence of viscous resistance.

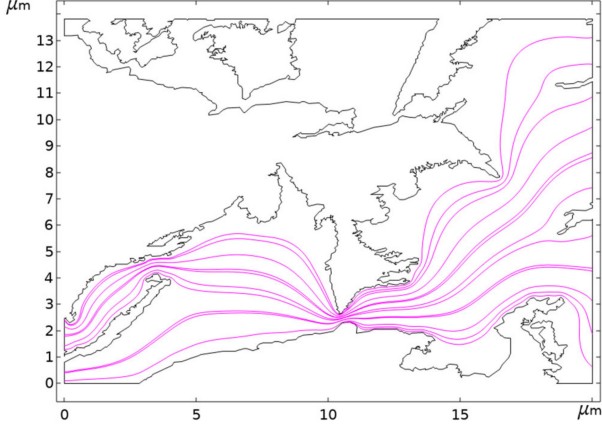

**Figure 20.** Flow line distribution of the flow field of the mesoscale fluid with an analytic structure.

To clearly grasp the flow situation and distinguish the flow areas from the non-flow areas, surface and isoline maps reflecting the pressure distribution in the pore space were plotted (Figure 21 where the areas without marked pressure isolines are the non-flow areas).

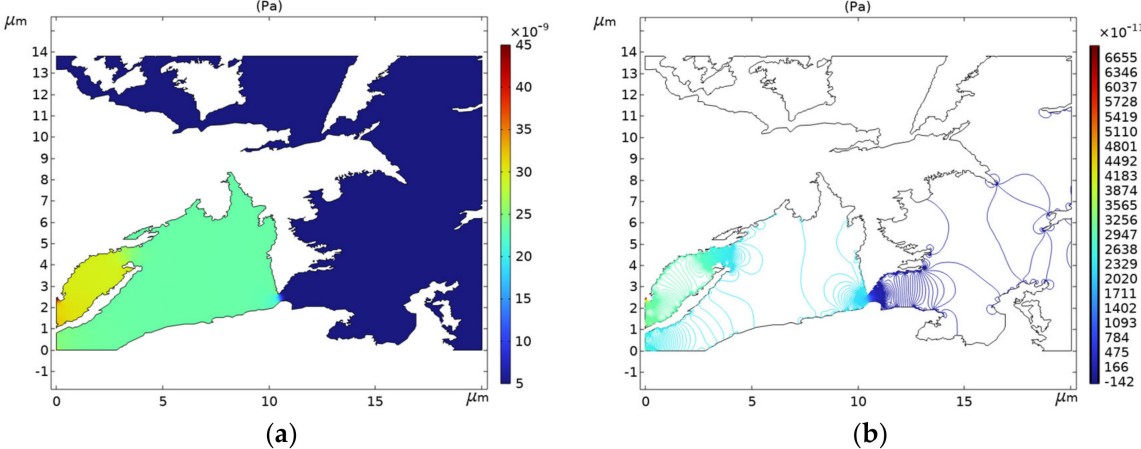

**Figure 21.** Isoline map of flow pressure field of mesoscale fluid with analytical structure. (**a**) Surface map; (**b**) Isoline map.

In the isoline map, after the fluid flows through the "throat" of the pore structure at x = 10–15 μm, a sudden pressure drop can be observed. Such a pressure drop occurs mainly because massive energy is consumed to pass through the small-sized "throat" here.

## 4. Application and Guidance for Engineering

As illustrated in Figure 21, both non-flow areas and flow areas exist in the pore space where the fluid flows. This finding can explain many engineering phenomena.

### 4.1. Project Background and Overview

In the pipes of the methane extraction system of a high-methane coal mine where the author has worked, and as one of the main index gases of coal spontaneous combustion, CO gas has often been detected, as a concentration ranging from about 2 to 20 ppm, from the pre-pumping boreholes in a coal seam. This concentration does not exhibit a continuous upward trend over a long period, which has greatly disturbed the early prediction and early warning of coal spontaneous combustion. According to the previous research, the CO gas comes from two possible sources, i.e., the gas occurring in existing coal seams [39] and the gas generated by coal oxidation. The specific sources of coal oxidation may be the coal debris remaining in the borehole during the drilling process, or the broken coal body area through which the borehole passes.

### 4.2. Verification of the First Source

The authenticity of the first source was verified through a combination of laboratory analysis and underground field experiments.

(1) Laboratory analysis

In laboratory analysis, first, fresh coal blocks were collected underground with their outer pre-oxidized layers stripped in inert gas, crushed, and put in a coal sample tank where they were allowed to desorb gases in a natural state (Table 4). After 24 h, the components and concentrations of desorbed gases were analyzed by a gas chromatograph every 4 h. The concentration of CO was found to be 0 ppm (Table 5), which means that the first source is ruled out in laboratory analysis.

**Table 4.** Composition and concentrations of desorbed gases.

| N$_2$ | CH$_4$ | CO$_2$ | C$^0_2$~C$^0_8$ |
|---|---|---|---|
| 37.49% | 59.56% | 2.92% | 0.03% |

**Table 5.** Contents of original CO gas recorded in the laboratory test.

| No. | Time (h) | CO (ppm) | O$_2$ (%) |
|---|---|---|---|
| 1 | 0 | 0 | 0 |
| 2 | 4 | 0 | 0 |
| 3 | 8 | 0 | 0 |
| 4 | 12 | 0 | 0 |
| 5 | 16 | 0 | 0 |
| 6 | 20 | 0 | 0 |
| 7 | 24 | 0 | 0 |
| 8 | 28 | 0 | 0 |
| 9 | 32 | 0 | 0 |
| 10 | 36 | 0 | 0 |
| 11 | 40 | 0 | 0 |
| 12 | 44 | 0 | 0 |
| 13 | 48 | 0 | 0 |

(2)  Underground field experiment

In the underground field experiment, an upward borehole (with an inclination of about 10°) was drilled using a drilling machine at the intact coal wall exposed at the working face. Then, two copper tubes were placed in the borehole. Tube A reached the bottom of the borehole, tube B reached the middle, and the two copper tubes were fixed with sealing materials at the top of the borehole (Figure 22). The borehole was 30 m in depth, 10 m of which was sealed from the top.

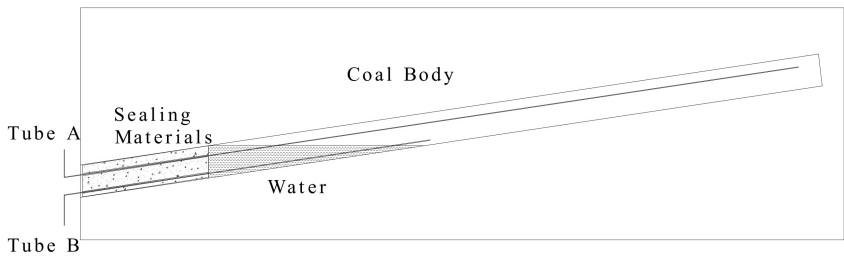

**Figure 22.** Schematic diagram of drilling construction.

**Water sealing:** To test and improve the sealing effect, the borehole needed to be sealed with water. First, the water was slowly injected into tube A. As the water rose from the bottom of the borehole and overflowed tube B, water injection ceased and the water sealing process was completed. If water still leaked from the borehole after sealing, the borehole needed to be resealed with sealing materials.

**Borehole washing:** During drilling, a large amount of CO was generated as the drill pipe rubbed against the inner wall and then remained in the borehole after the drill pipe's withdrawal. Therefore, the borehole needed to be washed with inert gas. First, N$_2$ was slowly injected into tube A. Afterwards, the components and concentrations of the discharged gas were detected at the outlet of tube B. When the discharged gas no

longer contained CO anymore, $N_2$ injection stopped and the borehole washing process was completed. After borehole washing, tubes A and B were sealed.

**Gas monitoring:** After the copper tubes were sealed for 10 days, the gas samples were taken from tube A and sent to the gas chromatograph to detect their components and concentrations for five consecutive days. During gas sampling, tube B was connected to the $N_2$-filled gas bag before gas extraction from tube A, which could prevent the outside gas in the roadway from interfering with gas components in the borehole.

According to the detected underground field experimental results of the gas components and concentrations measured for five consecutive days, the CO concentration was 0 ppm (Table 6), which proves that the gas originally occurring in the coal seam does not contain CO. That is, the possibility of the first source is excluded.

**Table 6.** Experimental data of field measurement of the original CO content.

| Date | $N_2$ (%) | $O_2$ (%) | CO (ppm) | $CO_2$ (%) | $CH_4$ (%) | $C_2H_6$ (%) | $C_2H_4$ (%) | $C_2H_2$ (%) |
|---|---|---|---|---|---|---|---|---|
| 12 July 2021 | 80.0295 | 4.4236 | 0 | 0.4332 | 15.1123 | 0.0014 | 0 | 0 |
| 13 July 2021 | 79.8978 | 4.3658 | 0 | 0.7898 | 14.9334 | 0.0132 | 0 | 0 |
| 14 July 2021 | 81.8217 | 4.6952 | 0 | 1.7355 | 11.745 | 0.0026 | 0 | 0 |
| 15 July 2021 | 83.0248 | 4.5234 | 0 | 1.4566 | 10.9881 | 0.0071 | 0 | 0 |
| 16 July 2021 | 82.4895 | 4.6533 | 0 | 1.5253 | 11.3232 | 0.0087 | 0 | 0 |

*4.3. Verification of the Second Source*

For the second source, this study holds the view that coal oxidation needs to go through three stages in sequence, i.e., physical adsorption, chemical adsorption, and chemical reaction. This mainly arises from the intermolecular forces and the energy required for the reaction. In the methane extraction borehole, gas in the microscale pore structure of the coal body is distributed in both the flow and non-flow areas under the action of negative extraction pressure.

In the non-flow area of the microscale pore structure, the diffusing oxygen molecules are able to adsorb and react with the active functional groups on the coal surface at the gas–solid interface. Under the concentration gradient, the generated CO gas will diffuse into the flow area in the microscale pore space and then enter the macroscale borehole and extraction pipe under the negative extraction pressure. Finally, the CO gas will be detected and analyzed by the gas chromatograph. As the reaction between oxygen and the active functional groups at the gas–solid interface continues, the oxygen molecules in the non-flow area are consumed constantly until they cannot support the oxidation reaction anymore. Consequently, the reaction ceases. Oxygen molecules in the flow area in the microscale pore structure are subject to both Darcy's law of seepage and Fick's law of diffusion, but their movement is dominated by the former. Due to the limited Fick's law of diffusion, only a very small amount of oxygen molecules can flow into the non-flow area to support a slow oxidation reaction. Therefore, the CO concentration in the extraction pipe does not exhibit a continuous increase.

On the other hand, the $O_2$ gas concentration in the extraction borehole is about 10%. These oxygen molecules will react with the reactive functional groups on the surface of the drilling debris in the borehole or on that of the coal body in the fractured area. CO is produced during this reaction. However, heat is unlikely to accumulate heat in the borehole due to the high gas flow velocity there. Therefore, when the reaction of the reactive functional groups finishes, the accumulated heat is insufficient to provide the activation energy required for the reaction between oxygen molecules and those less active functional groups. Resultantly, the reaction phases out. This is another reason why the CO concentration in the extraction pipe does not exhibit a continuous increase.

*4.4. Summary*

In summary, the CO gas in the methane extraction pipe of the coal seam pre-extraction borehole comesderives from two sources, i.e., the macroscale outer surface of the coal body and the microscale non-flow area in the pore structure. In the absence of sufficient heat, the CO gas originating from the macroscale outer surface of the coal body will gradually stop reacting. In contrast, the CO gas originating from the microscale non-flow area in the pore structure continues to react slowly. Under the joint effect of the two discontinuous sources, the CO gas concentration in the methane extraction pipe of the pre-extraction borehole does not exhibit a continuously increasing trend.

## 5. Conclusions

In this paper, the pore structure characteristics of coal were characterized at the mesoscale with the help of a pore size and specific surface analyzer and an SEM. Moreover, a numerical simulation study on the fluid flow in pores was carried out using the cross-scale method on the basis of digital reconstruction. The main conclusions are as follows:

(1) Influenced by pore structure characteristics and fluid viscosity, not all fluids in the pore space are flowing. The non-flow area is mainly located in small-sized pores.
(2) The order of magnitude of the macroscale flow velocity of the gas flow field in goafs is "m/s", while that of the mesoscale flow velocity of the gas flow field in pores is "nm/s".
(3) A generation pattern of CO gas has been revealed. It mainly originates from the mesoscale gas–solid interface in the non-flow area in the pore structure, and well explains the source of CO gas in engineering practice and the discontinuous growth phenomenon in the methane extraction pipe.

**Author Contributions:** B.D.: Conceptualization, methodology, validation, formal analysis, writing-original draft, visualization. Y.L.: Methodology, writing—review & editing, supervision, project administration, funding acquisition. F.T.: Writing—review & editing, supervision, project administration, funding acquisition. B.G.: Methodology, data curation. All authors have read and agreed to the published version of the manuscript.

**Funding:** This research was funded by the National Natural Science Foundation of China (grant no. 52174229,52174230) and the China Postdoctoral Science Foundation (grant no. 2021MD703848).

**Institutional Review Board Statement:** Not applicable.

**Informed Consent Statement:** Not applicable.

**Data Availability Statement:** The experimental data can be provided upon request to the corresponding author.

**Acknowledgments:** The authors are grateful for the financial support received from the National Natural Science Foundation of China (grant no. 52174229,52174230) and the China Postdoctoral Science Foundation (grant no. 2021MD703848). The authors also thank the reviewers and editors for the valuable advice and comments on the articles.

**Conflicts of Interest:** The authors declare no conflict of interest.

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
