# Peer review of "Analytical Prediction of Coal Spontaneous Combustion Tendency: Pore Structure and Air Permeability"

_sustainability, doi:10.3390/su15054332_

Round 1
Reviewer 1 Report
The manuscript focuses on coal spontaneous combustion and puts an emphasis on the meso-scale effects of gas flow field in goafs. On the whole, the structure of this manuscript is clear, and the experiment and numerical simulation are described neatly.
(1)However, the presentation of Introduction section in the manuscript is not smooth. It is recommended to rewrite it.
(2)According to the literature review, what is macro-scale and micro-scale effects of gas flow field effects on coal spontaneous combustion ?
(3)This manuscript used SEM images as a meso-scale structural reconstruction method, what is the strengths and weaknesses of this method compared with 3D Micro-CT ?
(4)During coal spontaneous combustion, how does temperature increase influence the fluid flow in porous medium?
(5)How does the pressure distribution influence the moisture transportation in porous medium?
(6)There are papers that I have reviewed in the past years. Please consider the suggested research in your paper when enhancing the literature review. I believe they are worth considering in your paper.
Zhang, X.; Pan, Y. Preparation, Properties and Application of Gel Materials for Coal Gangue Control. Energies 2022, 15, 557. https://doi.org/10.3390/en15020557
Zhang X, Zhou F, Zou J. Numerical Simulation of Gas Extraction in Coal Seam Strengthened by Static Blasting, Sustainability 2022, 14(19), 12484; https://doi.org/10.3390/su141912484
Jiaxing Zou, Rui Zhang, Fengyuan Zhou, Xiaoqiang Zhang. Hazardous area reconstruction and law analysis of coal spontaneous combustion and gas coupling disasters in goaf based on DEM-CFD, https://doi.10.1021/acsomega.2c07236
Author Response
Response to Reviewer 1 Comments
Point 1::The presentation of Introduction section in the manuscript is not smooth. It is recommended to rewrite it.
Response 1:According to the reviewer’s comments, we have rewritten the Introduction section. As shown in the modified and resubmitted manuscript, the Introduction section has been rewritten in blue characters.
Point 2:According to the literature review, what is macro-scale and micro-scale effects of gas flow field effects on coal spontaneous combustion ?
Response 2:In the micro-scale aspect, the gas flow field can provide a continuous supply of fresh air for the coal-oxygen reaction. The oxygen molecules in the fresh air can react with molecules or active functional groups on the pore structural surface of coal,which will release large amount of reaction heat. In the macro-scale aspect, the effects of air flow field mainly affect the coal temperature. The fast air flow can take away the heat stored in the coal body,which will reduce the temperature of the coal body.
Point 3:This manuscript used SEM images as a meso-scale structural reconstruction method, what is the strengths and weaknesses of this method compared with 3D Micro-CT ?
Response 3:The method used in the manuscript for pore structural reconstruction mainly depends on the acquired SEM images. Therefore, the reconstructed pore structure represents an interface or a plane in three-dimensional flow space. Compared with 3D Micro-CT, the strengths of this method is that it can save computing resources, shorten modeling time and computing time, and at the same time, it does not reduce computing accuracy.
Point 4:During coal spontaneous combustion, how does temperature increase influence the fluid flow in porous medium?
Response 4:When the temperature rises, the thermal movement of fluid molecules in porous media intensifies, and at the same time, the fluid density decreases due to thermal expansion and the flow accelerates. It shows that the increase of temperature promotes the gas flow. At the same time, the temperature rise will also cause the matrix to expand and compress the pore space, which indicates that the temperature rise hinders the gas flow.
Point 5:How does the pressure distribution influence the moisture transportation in porous medium?
Response 5:The pressure distribution in the pores will directly act on the moisture surface, causing moisture flow. On the other hand, the gas in the pores will flow from the place with high pressure to the place with low pressure, which will carry the evaporated moisture to flow in the pores.
Point 6:There are papers that I have reviewed in the past years. Please consider the suggested research in your paper when enhancing the literature review. I believe they are worth considering in your paper.
Response 6:The articles have been taken into consideration when revising the references. What is more, combining with other reviewers’ comments and suggestions, we have added a reference whose series number is 12 to enrich our manuscript with international research and demonstrate the depth of our manuscript.
Please see the attached manuscript.

Reviewer 2 Report
The authors have investigated Mesoscale Study of Gas Flow Effects on Coal Spontaneous Combustion. My suggestions are given below:
1) The novelty of the work needs to be clearly stated.
2) List the assumptions used in numerical study.
3) How is the convergence criteria defined for the simulation in COMSOL?
4) Is the numerical study performed with transient assumption?
5) Is the gas velocity field contour in Fig.3-4 time averaged?
Author Response
Response to Reviewer 2 Comments
Point 1: The novelty of the work needs to be clearly stated.
Response 1: In previous researches, many scientists and researchers have carried out related study about coal spontaneous combustion in microscale and macroscale. But the macro-scale study of coal clusters and piles can not reveal the nature of oxidation and combustion, and the micro-scale study of coal molecule and functional groups can not be directly applied to engineering practice. According to the literature survey, coal is a porous medium and its spontaneous combustion is a multi-scale process. Thus, meso-scale study of coal spontaneous combustion, especially the pore structural characteristic and permability, becomes essential and crucial. In this manuscript, meso-scale structure in coal body ( such as pore size, pore volume, and specific surface area), and meso-scale structure morphological characteristics on coal surface have been finely analyzed and characterized. On this basis, meso-scale structure of pore and fracture was digitally reconstructed. What is more, velocity and pressure distribution of flow field in pores on the scanning plane was outlined and described by numerical simulation, which indicated that not all the fluids in pores were flowing because of the pore structure characteristics and fluid viscosity. This conclusion analytically predicted coal spontaneous combustion tendency and well explained the source of CO gas in the methane extraction pipes, which is one of the main index/indicator gases of coal spontaneous combustion.
Point 2: List the assumptions used in numerical study.
Response 2:
- The key parameters used in the numerical study is shown in Table 3-1 of the modified manuscript.
Table 3-1 Numerical simulation parameters
|
Parameter |
Value |
Description |
Unit |
Parameter |
Value |
Description |
Unit |
|
|
29 |
Molar mass of air |
- |
|
5.38e-6 |
Permeability in the natural accumulation area |
m2 |
|
|
0.29 |
Void ratio of goaf |
- |
|
2.6e-6 |
Permeability in the load-affected zone |
m2 |
|
|
2.01e-5 |
Dynamic viscosity |
Pa·s |
|
1.3e-6 |
Permeability in the compacted stable zone |
m2 |
|
|
2.88e-2 |
Gas diffusion coefficient |
- |
|
300*180 |
Goaf area |
m2 |
- In Fig.3-2, the airflow velocity at the inlet of the goaf was set to 0.9 m/s.
- In Fig.3-4, The porosity of the coal block was 9.5% and the permeability was 1e-16 m2. The left boundary of the computational domain is the inflow boundary, and it is set that the airflow velocity is 0.04m/s.
Point 3: How is the convergence criteria defined for the simulation in COMSOL?
Response 3:
The relative tolerance is set to 0.001. The error estimation factor is set to 1.The rest parameter is set to default values.
Point 4: Is the numerical study performed with transient assumption?
Response 4: The numerical simulation study is performed with the steady-state assumption.
Point 5: Is the gas velocity field contour in Fig.3-4 time averaged?
Response 5: The Fig.3-4 indicates the distribution of gas velocity field. It is a calculated result when the flow is in a steady state.

Reviewer 3 Report
The manuscript "Mesoscale Study of Gas Flow Effects on Coal Spontaneous Combustion in Goafs: A Numerical Simulation and Experimental Study" by Bin Dua, Yuntao Lianga, Fuchao Tiana and Baolong Guob was submitted for peer review.
I read the submitted manuscript with great interest. The authors turned to a very urgent problem: study of coal structure and the influence of the structure (the pores size and the presence of moisture) on air permeability and velocity.
A great deal of research has been done by the authors. But the manuscript has significant flaws that need to be corrected.
From my point of view, there is no single vector in the manuscript. The title of the manuscript does not fully reflect the essence of the study and the final result.
Correction of the shortcomings listed below must be done to improve the quality of the manuscript, enhance the ease of perception of the presented material and increase the interest of a readers.
1.) From my point of view, the title of the manuscript is a bit vague and does not reflect the essence of the study. The authors solve a rather urgent problem: to establish the possibility of spontaneous combustion of coal depending on the size of coal’s pore opening and the permeability of air through these pores. However, there is the phrase: ‘of Gas Flow Effects’ in the title, which is slightly misleading, since it can be understood that gas flow in the mine is being considered. Also, from my point of view, the phrase: ‘A Numerical Simulation and Experimental Study’ is redundant since these are research methods and belong to the Materials and Methods section. The title must indicate the essence of the study, and not the way to achieve the goal of the study.
2.) From my point of view, there are very few keywords. Keywords enable the reader to quickly search for the necessary material and enable the author to popularize their research and increase interest and citations. But if this number of keywords satisfies the requirement of the journal, this comment is advisory.
3.) The abstract is not quite formed correctly. It is very blurry and framed incorrectly. Abstract is a concise and succinct presentation of a complicated study. It seems that the authors have taken certain phrases from the text and thus formed the abstract. The abstract should clearly indicate the purpose of the study, its importance for society (i.e. to characterize the problem), identify the methods and materials of the study, and the conclusions should be clearly and briefly formulated. There is no "starting point" in the abstract, that is, information about previous studies (one sentence is enough). From my point of view, in the abstract, such information begins with the statement: "Previously conducted studies have established that ...".
3.1) It is desirable to avoid narrative text in the abstract.
3.2) Try to use words and phrases: an analysis has been carried out; studied; developed; proposed; established and so on. It is advisable to start sentences in the abstract with these words and phrases.
3.3) At the end of the abstract, it is necessary to indicate the final result obtained by the authors, for example: A model has been developed that allows ...; A dependence has been established which is...; A pattern has been revealed...; An efficient system (technology) has been proposed, and so on.
The abstract should be revised.
4.) The manuscript has a sufficient list of references (38 references in total). At the same time, there is no comprehensive coverage of research in terms of geography of citations. There are not enough references to international experience in the field, no more that 5-6 sources in total, this is very few. The list of references is intended to demonstrate the depth of the authors' study, the relevance of the material and interest of their research.
4.1.) The depth of study is demonstrated with the number of references – is on the verge of sufficiency.
4.2.) Relevance – with the availability of research in recent years – is enough.
4.3.) Interest – with the availability of research by scientists from different countries - is not enough.
Since you are publishing your manuscript in an international publication, it is necessary to demonstrate the international relevance and interest of this issue. This can be done by analyzing the studies of scientists from different countries. It is imperative to supplement the list of references with studies of scientists from different countries over the past 3-5 years to show geographical (general/global) interest and relevance.
The List of References needs to be completed.
5.) It is necessary to avoid group references: [1-4] or [12-15]. Each paper you refer is unique and the studies you refer deserve more proper and careful review to demonstrate (and prove) its importance for the current research. It is necessary to demonstrate in detail the essence of each study and their need for your work.
6.) From my point of view, the authors abuse the names of scientists when mentioning the study Wang et al. (line 50), Yang and Li (line 52) and so on. The authors indicate the name of the researcher (or group of researchers), then indicate their achievement, after which they make a reference to the study. From my point of view, references [9], [10] are sufficient without mentioning the surname at the beginning of the sentence. If the reader is interested in the name of the researcher, then it is easy to refer to the references list. It is important for the reader to know the essence (main idea) of the disclosed issue, not the name of the researcher.
7.) From my point of view, at the end of the introduction, there is no brief conclusion of earlier papers. The authors did not summarize their analysis and did not identify unresolved issues. This conclusion should make it possible to characterize the actual question posed, the purpose of the study and the tasks to be solved to achieve this goal. For example: Analyzing the above, it can be noted that ... is a very topical issue. Therefore, the purpose of this study is ... and to achieve this, it is necessary to solve the following tasks: 1); 2); ...
Such a conclusion allows the authors to correctly formulate the conclusions and the reader to understand the vector of the study. It needs to be improved.
8.) When analyzing previous studies, the authors make statements that are not supported by evidence (references). Many statements are very broad and very difficult to understand. From my point of view, it is necessary to form more compact sentences, this way you avoid group references.
9.) Considering the comments (3), (4), (6) and (8), I would like to note that the authors have very poorly disclosed the main subject of the study.
In recent years, a lot of work has been carried out to study the safety of mining operations. Spontaneous combustion depends not only on the ability of coal dust to absorb oxygen, but also on the concentration of this dust. A lot of work has been devoted to the study of dust concentration in certain areas and the development of technologies that minimize such concentration.
For example,
9.1) Golik, V.I., Stas, G.V., Liskova, M.Yu., Kongar-Syuryun, C.B. Improvement of the occupational safety by radical isolation of pollution sources during underground ore mining. Bezopasnost' Truda v Promyshlennosti 2021, 2021(7), 7–12. https://doi.org/10.24000/0409-2961-2021-7-7-12
9.2) Golik, V.I., Gashimova, Z.A., Liskova, M.Yu., Kongar-Syuryun, Ch.B. To the problem of minimizing the volume of mobile dust in the development of pits. Bezopasnost' Truda v Promyshlennosti 2021, 2021(11), 28-33. https://doi.org/10.24000/0409-2961-2021-11-28-33
As follows from the presented works (9.1), (9.2) the authors of the manuscript submitted for review missed a large layer of research related to the impact of underground structures on rock mass. If the authors become familiar with the works presented in (9.1), (9.2) they will be able to properly form the introduction, enrich their manuscript with international research and demonstrate the depth of their material, as well as eliminate the remark (3).
10.) The authors did not disclose the material used in the study: coal dust. From my point of view, it is necessary to specify the place (enterprise) of material selection, indicate its characteristics and composition. Such information will be of interest to the readers if they want to repeat the experiment.
11.) In the conclusion section, point (3) is very vague.
Summary: The manuscript is not a finished research work. The corrections are needed. The chosen research topic is relevant. From my point of view, the authors failed to present their research correctly and clearly, which reduced its value and worsened the ease of perception of the material presented.
From my point of view, the manuscript cannot be published in the open press without correction in accordance with my suggestions.
Author Response
Response to Reviewer 3 Comments
Point 1: From my point of view, the title of the manuscript is a bit vague and does not reflect the essence of the study.
Response 1:
According to reviewer’s comments, it has been modified with “Analytical Prediction of Coal Spontaneous Combustion Tendency: Pore Structure and Air Permeability” in blue characters.
Point 2: From my point of view, there are very few keywords. Keywords enable the reader to quickly search for the necessary material and enable the author to popularize their research and increase interest and citations. But if this number of keywords satisfies the requirement of the journal, this comment is advisory.
Response 2:
Thanks for the reviewer’s advisory comment. The keywords have been changed into: “Mesoscale; Pore structure; Fire risk assessment; Coal spontaneous combustion; Ignition tendency”
Point 3: The abstract is not quite formed correctly. It is very blurry and framed incorrectly.
Response 3:
According to the reviewer’s comments, we have rewritten the Abstract section. As shown in the modified and resubmitted manuscript, the Abstract section has been rewritten in blue characters.
Point 4: The manuscript has a sufficient list of references (38 references in total). At the same time, there is no comprehensive coverage of research in terms of geography of citations. There are not enough references to international experience in the field, no more that 5-6 sources in total, this is very few. The list of references is intended to demonstrate the depth of the authors' study, the relevance of the material and interest of their research.
Response 4: Combining with the reviewer’s comments — point 9, we have added closely related article in the references whose series number is 12.
Point 5: It is necessary to avoid group references: [1-4] or [12-15]. Each paper you refer is unique and the studies you refer deserve more proper and careful review to demonstrate (and prove) its importance for the current research. It is necessary to demonstrate in detail the essence of each study and their need for your work.
Response 5:
As far as I am concerned, the references of [1]-[4] respectively reported the spontaneous combustion of coal in different countries around the world. These types of coal fires are not completely the same. Some are caused by environmental factors, and some are caused by human beings. Some are coal field fires on the surface, and some are underground fires. However, the types of coal fires are not completely consistent with the theme of this paper, because the theme of this paper is not to study different types of coal fires. Therefore, we adopted the method of group reference in the manuscript rather than detailed introduction about these articles.
In my point of view, the references of [12]-[15] were published by one author, and all of them used the research method of model compounds to study active functional groups in different problems. Therefore,we adopted group reference in this paper .
Point 6:From my point of view, the authors abuse the names of scientists when mentioning the study Wang et al. (line 50), Yang and Li (line 52) and so on. The authors indicate the name of the researcher (or group of researchers), then indicate their achievement, after which they make a reference to the study. From my point of view, references [9], [10] are sufficient without mentioning the surname at the beginning of the sentence. If the reader is interested in the name of the researcher, then it is easy to refer to the references list. It is important for the reader to know the essence (main idea) of the disclosed issue, not the name of the researcher.
Response 6:In the modified and resubmitted manuscript, we have changed and rewritten the Introduction section.We adopt new expressions and focus on the key research content.
Point 7:From my point of view, at the end of the introduction, there is no brief conclusion of earlier papers. The authors did not summarize their analysis and did not identify unresolved issues. This conclusion should make it possible to characterize the actual question posed, the purpose of the study and the tasks to be solved to achieve this goal.
Response 7:In the modified and revised manuscript, we have rewritten the Introduction section. At the end of the Introduction section, we have made a clear and brief conclusion with a series of compact sentences. These sentences identified unsolved issues and proposed tasks to be solved. The rewritten Introduction section has been marked in blue characters.
Point 8:When analyzing previous studies, the authors make statements that are not supported by evidence (references). Many statements are very broad and very difficult to understand. From my point of view, it is necessary to form more compact sentences, this way you avoid group references.
Response 8:In the Introduction section, we have changed expression style. To make statements clear and comprehensible, we have changed sentences into tables, and indicated the source of evidence (references), as seen in Table 1-1.
Point 9: If the authors become familiar with the works presented in (9.1), (9.2) they will be able to properly form the introduction, enrich their manuscript with international research and demonstrate the depth of their material, as well as eliminate the remark (3).
Response 9: Thanks for the reviewer’s constructive suggestions. When we modify and revise the Introduction Section and Abstract Section, we carefully read the articles proposed by (9.1) and (9.2). In addition, in order to enrich our manuscript with international research and demonstrate the depth of our manuscript, as well as eliminate the remark (3), we added a reference whose series number is 12, combining the comments of other reviewers.
Point 10:The authors did not disclose the material used in the study: coal dust. From my point of view, it is necessary to specify the place (enterprise) of material selection, indicate its characteristics and composition. Such information will be of interest to the readers if they want to repeat the experiment.
Response 10:In accordance with the reviewer’s suggestion and comment, we have added related information about coal sample in section 2.1.1 with blue characters.
Point 11: In the conclusion section, point (3) is very vague.
Response 11: Thanks for the reviewer’s advisory comments. In the modified and revised manuscript, we have modified point (3) with blue characters.
Round 2
Reviewer 3 Report
The manuscript "Analytical Prediction of Coal Spontaneous Combustion Tendency: Pore Structure and Air Permeability" by Bin Dua, Yuntao Lianga, Fuchao Tiana and Baolong Guob was submitted for second review.
As can be seen from the submitted manuscript and the explanatory note to the review, the authors did a lot of work to make changes in accordance with the comments.
The revised manuscript is a completed scientific study on a highly relevant topic: study of coal structure and the influence of the structure (the pores size and the presence of moisture) on air permeability and velocity. The revised version of the manuscript, in my opinion, fully satisfies the requirements of a scientific article and can be published in the open press.